# Design, Preparation and Evaluation of Supramolecular Complexes with Curcumin for Enhanced Cytotoxicity in Breast Cancer Cell Lines

**DOI:** 10.3390/pharmaceutics14112283

**Published:** 2022-10-25

**Authors:** Hamdy Abdelkader, Adel Al Fatease, Zeinab Fathalla, Mai E. Shoman, Heba A. Abou-Taleb, Mohammed A. S. Abourehab

**Affiliations:** 1Department of Pharmaceutics, College of Pharmacy, King Khalid University, Abha 62529, Saudi Arabia; 2Department of Pharmaceutics, Faculty of Pharmacy, Minia University, Minia 61519, Egypt; 3Department of Medicinal Chemistry, Faculty of Pharmacy, Minia University, Minia 61519, Egypt; 4Department of Pharmaceutics and Industrial Pharmacy, Faculty of Pharmacy, Merit University (MUE), Sohag 82755, Egypt; 5Department of Pharmaceutics, College of Pharmacy, Umm Al-Qura University, Makkah 21955, Saudi Arabia

**Keywords:** curcumin, cyclodextrins, hydroxyethyl β-cyclodextrin, MCF-7, solubility, cytotoxicity

## Abstract

Curcumin is one of the most researched phytochemicals by pharmacologists and formulation scientists to unleash its potential therapeutic benefits and tackle inherent biopharmaceutic problems. In this study, the native β-cyclodextrin (CD) and three derivatives, namely, Captisol (sulfobutyl ether β-CD), hydroxypropyl β-cyclodextrin, and hydroxyethyl β-cyclodextrin were investigated for inclusion complexes with curcumin using two preparation methods (physical mixing and solvent evaporation). The prepared complexes were studied for docking, solubility, FTIR, DSC, XRD, and dissolution rates. The best-fitting curcumin: cyclodextrins (the latter of the two CDs) were evaluated for cytotoxicity using human breast cell lines (MCF-7). Dose-dependent cytotoxicity was recorded as IC50% for curcumin, curcumin: hydroxyethyl β-cyclodextrin, and curcumin: hydroxypropyl β-cyclodextrin were 7.33, 7.28, and 19.05 µg/mL, respectively. These research findings indicate a protective role for the curcumin: hydroxypropyl β-cyclodextrin complex on the direct cell lines of MCF-7.

## 1. Introduction

Despite the appreciable advances in anticancer treatment modalities, cancer alarmingly increases among the world population and remains a major cause of death worldwide [1]. An estimate was made in 2020 by the World Cancer Research Fund International (WCRFI) that the number of cancer cases involved 18 million cancer patients [2,3]. High toxicity, poor tolerability by patients, and low cellular uptake by anticancer cells are among the causes of increased mortality from cancer.

Therefore, there has been growing interest over the past three decades to find out alternative anticancer treatments that can be more effective and less toxic; therefore, they can be used for prevention and/or treatment [1,4,5]. The search for more tolerable and less toxic anticancer agents could be a promising strategy for fighting cancer in the coming decades.

There have been several anticancer agents that have come from natural origins (plants, marine origins, and microbes), which could fit into this strategy [1,6]. Several anticancer substances have been extracted from plant origins, such as polyphenolic compounds (e.g., curcumin, resveratrol, and catechins) and flavonoids (e.g., rutin and quercetin) [5,7]. Mitomycin C, doxorubicin, and bleomycin originate from microbes, while those that come from marine origin include bryostatins, ecteinascidin 743, and kahalalide F [6].

Curcumin is a natural polyphenolic compound extracted from turmeric rhizomes [1]. Curcumin has attracted the attention of scientists for decades as a potential anticancer agent and a nutrient for the prevention of some common types of cancers, such as breast, brain, lung, and prostate cancers [1,8]. To the best-known knowledge, curcumin exerts its cytotoxic activities through apoptotic, anti-proliferative, and the inhibition of cellular signaling pathway mechanisms [9,10]. For example, curcumin’s role in breast cancer was reported to mediate an apoptotic mechanism in the breast cancer cells through the inhibition of NFk_B_, cyclin D, and MMP-1 [11]. Other adjuvant mechanisms of curcumin that have been reported for the prevention of cancers include antioxidants, anti-inflammatories, antiglycation, and metal-chelating properties [12,13]. The major challenges faced by the development of curcumin into nutraceutical products include poor water-solubility, oral bioavailability, and poor stability [1,10]. Several strategies have been attempted to enhance the water-solubility and stability of curcumin through its encapsulation in colloidal systems such as micelles, emulsions, liposomes, solid lipids, and polymeric nanoparticles [13,14]. The inclusion of a hydrophobic drug within the cyclodextrin cavities still remains one of the most successful techniques to enhance both the solubility and stability of several drug candidates [15,16,17]. This is for simplicity, a large solubilization capacity, approval by regulatory bodies, and low toxicity [16,17]. For example, the curcumin: β-cyclodextrin complex was prepared by the saturated aqueous solution technique. This complex showed superior solubility and anticancer activities compared to free curcumin [18]. In another study, the curcumin: β-cyclodextrin complex showed enhanced anticancer activities in prostate cancer cell lines [19]. In this study, four different cyclodextrin derivatives were investigated with curcumin in a 1:1 molar ratio rather than using a higher ratio of cyclodextrins in order to eliminate any potential irritation and toxicity due to the excessive use of excipients [17]. The four different cyclodextrins were β-cyclodextrin, hydroxyl propyl β-cyclodextrin, sulfobutyl ether β-cyclodextrin (Captisol^®^), and hydroxyl ethyl β-cyclodextrin. These four cyclodextrins were employed to inform about the best-fitting cyclodextrin that could enhance both the solubility and anticancer activities in human breast cancer cell lines. Two different techniques were used to prepare curcumin-cyclodextrins complexes, namely, the physical mixing and coprecipitation techniques. The prepared mixtures were investigated for solubility, dissolution, docking, thermal and spectral analyses, and in vitro evaluation using human breast cell lines.

## 2. Materials and Methods

### 2.1. Materials

Curcumin 97% was purchased from Alfa Aesar, Thermo Fisher Scientific, Heysham, Lancashire, UK. β-cyclodextrin, hydroxypropyl β-cyclodextrin, and hydroxyethyl β-cyclodextrin were purchased from Acros Organics, NJ, USA. Captisol (β-cyclodextrin sulfobutyl ether sodium salt) was a generous gift from CyDex Pharmaceuticals, Lawrence, KS, USA.

### 2.2. Preparation of Curcumin:Cyclodextrin Physical and Coprecipitated Mixtures

Curcumin: β-cyclodextrin, Captisol, hydroxypropyl β-cyclodextrin, and hydroxyethyl β-cyclodextrin physical mixtures were prepared by weighing the equivalent in weight (mg) of molar ratio separately. Drug:CDs were mixed thoroughly in a porcelain dish for 5 min using a spatula and sieved through a 250-µm sieve. 

For coprecipitated dispersion, curcumin was dissolved in methanol (50 mL), and CDs were dissolved individually in distilled water (10 mL). The methanolic solution of curcumin and the aqueous solutions of the CDs were mixed in a larger porcelain dish (100 mL capacity). The porcelain dish was placed on a hot plate stirrer, LabTech, Daihan, Seoul, Korea, adjusted to 80 °C, and left until the complete evaporation and casting of the solid dispersion. The casted powder was pulverized using a pestle and sieved through a 250 µm sieve.

### 2.3. Characterization of Curcumin: CD Mixtures

#### 2.3.1. Molecular Docking

Molecular Operating Environment (MOE) 2014.09 software (Chemical Computing Group, Montreal, QC, Canada) was used to execute molecular docking investigations to anticipate both the stability and potential orientation of curcumin inside the cavity and/or rim of various cyclodextrins; β-CD, HE β-CD, HP β-CD, and SBE β-CD (Captisol). The 3D structure of β-CD was downloaded from the Protein Data Bank (PDB) at https://www.rcsb.org (accessed on 1 March 2022) as the PDB file code: 5E6Z [20]. The 3D structure of HE β-CD, HP β-CD, and SBE β-CD were built using the builder interface of MOE software via substituting the hydroxyl group with ethyl, isopropyl, and sulfobuteryl radicals, respectively [21]. Hydrogens were added, and the energy of the CD structures was minimized to an RMSD (root mean square deviation) gradient of 0.01 kcal/mol using the QuickPrep tool of the MOE software. The compounds’ 3D structures were also built using a MOE builder. Curcumin was docked into the inclusion cavity of the various CDs using an induced-fit docking protocol, the Triangle Matcher method, and the dG scoring system for pose ranking. Following visual assessment of the resultant docking poses, poses with the highest stability and lowest binding free energy value were elected and reported.

#### 2.3.2. Solubility Studies

Equilibrium solubility studies were performed on the curcumin and curcumin: CDs in physical and dispersed (coprecipitated) mixtures using a water-bath shaker (Shel Lab water bath, Sheldon, Cornelius, OR, USA) at 37 °C ± 0.5 °C and a shaking rate of 120 strokes per minute (min). Excess amounts of samples were dispersed in individual conical flasks containing 10 mL of distilled water and were kept in the water bath for 24 h and left for another 24 h to attain equilibrium. An aliquot of 5 mL was withdrawn from the flasks and filtered, and the absorbance was measured by Shimadzu UV-2550 Spectrophotometer, Kyoto, Japan, at 470 nm.

#### 2.3.3. Differential Scanning Calorimetry (DSC)

Specific weights of curcumin and curcumin: CDs (β-CD, HP β-CD, HE β-CD, and Captisol) in the physical and dispersed mixtures were placed in a 40 µL-capacity aluminum pan and hermetically sealed and pierced. The pan temperature was gradually raised from 30 °C to 350 °C at 10 °C/min using a DSC calorimeter (DSC-60; Schimadzu, Kyoto, Japan). Nitrogen was used as a purging gas at a flow rate of 20 mL/min.

#### 2.3.4. Fourier Transform Infrared Spectroscopic (FTIR) Study

FTIR spectrophotometry (FT-IR, Tensor 37, Bruker, Billerica, MA, USA) was used to stack the spectra of curcumin and curcumin: CDs (β-CD, HP β-CD, HE β-CD, and Captisol) in the physical and dispersed mixtures. Spectra were collected from KBr discs. The spectra were collected in a range from 4000 to 400 cm^−1^.

#### 2.3.5. X-ray Diffraction (XRD)

The crystallinity of curcumin and curcumin: CDs (β-CD, HP β-CD, HE β-CD, and Captisol) in the physical and dispersed mixtures was studied. X-ray diffraction patterns were obtained by an XRD diffractometer (Unisantis XMD-300, GmbH, Germany). The tube was operated at 45,000 V and 0.8 mA. The scan range was 5–60° of a diffraction angle 2θ.

#### 2.3.6. In Vitro Dissolution Studies

Some selected curcumins and curcumin: CDs (HP β-, and HE β-CDs) in the physical and coprecipitated mixtures were studied for in vitro dissolution. Two dissolution media were investigated; the first dissolution medium was composed of simulated gastric fluid with pH 1.2, and the volume was 900 mL containing sodium lauryl sulphate (SLS) 1% w/w. The second dissolution medium was composed of simulated intestinal fluid (phosphate buffer with pH 6.8) containing 0.1% Tween 80. The volume of the dissolution media was 900 mL each; the media were paddled with the USP apparatus 2 at 100 rpm, and the temperature was set at 37 °C. Curcumin, PM, and Coppt mixtures weighing 20 mg or equivalent to 20 mg of curcumin were sprinkled onto the surface of the dissolution media. A sample of 5 mL was pipetted out at specified time points and replaced with another 5 mL of the fresh medium. The samples were analyzed spectrophotometrically, as previously mentioned.

#### 2.3.7. In Vitro Cytotoxicity Assay Using Human Breast MCF-7 Cells

Human breast MCF-7 epithelial cells (ATCC # HTB-22, ATCC, Manassas, VA, USA) were adopted in this study. Six different consecutive concentrations (3.125 to 100 µg/mL) of curcumin or its equivalent from curcumin: HP β-cyclodextrin and curcumin: HE β-cyclodextrin coprecipitated mixtures were prepared by dilution using the growth media. A 96-well plate was inoculated with 0.1 mL/well containing 1 × 10^5^ cells/mL incubated at 37 °C for 24 h [22]. The medium was removed from the 96-well plates, and the formed cell monolayer was washed twice with the media. Two-fold dilutions of the tested sample were made up using an RPMI medium with 2% serum [22]. Aliquots of 100 µL each of the diluted samples were added to 3 consecutive wells; the negative control was the blank medium. The plate was incubated at 37 °C and examined. The cells were examined microscopically for any signs of toxicity, a partial or complete loss of the monolayer, rounding, shrinkage, or cell granulation. An MTT solution was prepared (5 mg/mL in PBS) (Biobasic, Markham, Canada). A volume of 20 µL of MTT solution was added to each well and then placed in a thermostatically controlled shaker (ASAL 715 CT, Anzio, Italy) at 150 rpm for 5 min to thoroughly mix the MTT into the media and finally incubated (37 °C, 5% CO_2_) for 4 h. The media were discarded, and formazan (the MTT metabolic product) was dissolved in 200 µL DMSO by shaking at 150 rpm for 5 min. The optical density was measured using a microplate reader (mindray MR-96A, Shenzhen, China) at 560 nm and subtracted background at 620 nm.

### 2.4. Statistical Analysis

Statistical analysis was conducted using Graph Pad Prism software 8.4.3(686), San Diego, CA, USA, to test for the analysis of variance (ANOVA) with a statistical significance set at *p* < 0.05.

## 3. Results and Discussion

Four different curcumins: cyclodextrins (β-CD, HP β-CD, HE β-CD, and Captisol) mixtures were prepared using two techniques of physical mixing (PM) and the coprecipitation/solvent evaporation method. These two techniques are commonly used for the preparation of inclusion complexes of cyclodextrins and many poorly soluble drugs such as catechin, ketoconazole, meloxicam, and silymarin [17,23,24].

### 3.1. Docking Studies

The poses of curcumin docked into the inclusion pocket/cavity of β-cyclodextrin, HE-β-cyclodextrin, HP-β-cyclodextrin, and SBE-β-cyclodextrin (Captisol) are presented from three views: top view, bottom view and side view in order to mark the possible interactions and electrostatic/H–bond formation (Figure 1). Energy scores were also estimated, and the number of bonds is demonstrated in Table 1. The results indicated that both HE-β-cyclodextrin and HP-β-cyclodextrin showed more favorable interactions and a greater number of bonds and energy scores. More hydrophobic attractions were scored for the HE-β-cyclodextrin; this could indicate that the phenolic ring is more happily stable in the pocket of this cyclodextrin derivative.

### 3.2. Solubility Study

The solubility of curcumin alone and from the four different curcumin: cyclodextrin mixtures prepared by the two different methods: physical mixing (PM) and coprecipitation (Coppt), is presented in Figure 2. Curcumin solubility in water was very low at <2 µg/mL. Curcumin is a practically insoluble polyphenolic compound; similar results were reported elsewhere [24]. The results showed a significant (*p* < 0.05) enhancement of curcumin solubility irrespective of the method of preparation recorded for the four different complexes. The solubility of curcumin from the curcumin: cyclodextrin complexes ranged from 5 to 245 µg/mL; this was conducted in up to 2.5- to 122.5-fold increments. A superior enhancement was recorded for HE β-CD Coppt. The ranking of the solubility enhancement of curcumin was as follows: HE β-CD > HP β-CD > Captisol > β-CD. These results correlated well with the binding constants and docking poses recorded for the inclusion complexes. The higher the binding constant and the more stable the inclusion complex, the better the hiding of hydrophobic moieties in curcumin, and a greater solubility for curcumin was recorded [15,16]. The solubility from curcumin: cyclodextrin complexes was significantly (*p* < 0.05) dependent on the methods of preparation. For example, the solubility recorded for curcumin was 190 µg/ mL and 245 µg/ mL for curcumin: HE β -CD PM and HE β-CD Coppt, respectively. This could be ascribed to the fact that both curcumin and cyclodextrin were dissolved in methanol and water, respectively. This can generate more intimate interactions and a better complexation efficiency compared to the physical mixing of the two powders without processing.

### 3.3. DSC

DSC thermograms of curcumin and curcumin: cyclodextrin complexes were prepared using PM and Coppt techniques, as shown in Figure 3A–D. Curcumin demonstrated a thermal event at 185 °C due to the melting of curcumin crystals [24]. Broad peaks appeared from 50 to 120 °C ascribed for the four cyclodextrins. These broad peaks were due to bound moisture loss from the cyclic sugar moieties; the peaks were recorded from 250 to 300 °C due to the decomposition of cyclodextrins [24]. Curcumin: cyclodextrin, PM, and Coppt recorded weak peaks at a lower temperature compared to pure curcumin. These results indicate the interaction of the curcumin with cyclodextrin cavities and the formation of inclusion complexes. 

### 3.4. FTIR

Figure 4A–D presents the FTIR spectra of curcumin and curcumin: cyclodextrin complexes prepared by PM and Coppt. The FTIR spectra of cyclodextrins (cyclic sugars) showed a broad peak at 3300–3600 cm^−1^, indicating the stretching of the hydroxyl (-OH) groups and the stretching vibration of -CH from 1600 to1700 cm^−1^. Intense peaks of 2850 cm^−1^ were recorded for CDs due to C-H asymmetric/symmetric stretching.

The FTIR spectrum of curcumin showed characteristic bands for the phenolic OH groups that stretched at 3500 cm^−1^ [25]. Other sharp absorption bands at 1510 cm^−1^ and 1615 cm^−1^ were due to the stretching vibration of C=O and aromatic rings, respectively [26]. Curcumin’s characteristic bands disappeared from the FTIR spectra for curcumin: cyclodextrin complexes prepared by Coppt and PM, and the characteristic peaks belonging to the CDs were dominant. This indicates that the inclusion of hydrophobic aromatic rings in cyclodextrins cavities, hydrogen bonding, and Van der Waals forces formation [27].

### 3.5. XRD

The XRD spectra of curcumin and the four curcumin: cyclodextrin (β-CD, Captisol, HP β-CD, and HE β-CD) complexes are shown in Figure 5A–D. The curcumin XRD spectrum exhibited strong diffraction peaks, and this indicates a highly crystalline drug. Curcumin’s characteristic crystalline peaks were 2θ = 8.9°, 14.5°, 17.26°, 18°, 21.05°, 23.5°, 24.6°, and 28.2°. β-CD showed fewer characteristic crystalline peaks at 2.2°, 4.45°, and 15.6°.

With the exception of β-CD, all other three cyclodextrins (Captisol, HP β-CD, and HE β-CD) showed no diffraction peaks (very weak signals) or halo patterns; this indicates the complete amorphousness of these three cyclodextrins, as shown in Figure 5A–D. The four curcumin: cyclodextrins mixtures which were prepared using PM and Coppt, showed complete amorphousness and the disappearance of curcumin’s diffraction peaks.

### 3.6. In Vitro Dissolution in Simulated Gastric and Intestinal Fluids

The in vitro dissolution profiles of curcumin and the two selected curcumin: cyclodextrins, namely HP β-CD and HE β-CD, were prepared with the two techniques of PM and Coppt, as shown in Figure 6A,B. These two curcumin: cyclodextrin complexes were selected for showing the superior enhancement of curcumin’s solubility and high-binding constants as estimated from docking studies. The dissolution studies were performed in simulated gastric fluid (pH 1.2), Figure 6A, and simulated intestinal fluid (pH 6.8), shown in Figure 6B. For comparative purposes, two dissolution parameters Q_10min_ and Q_120min_ were obtained from the in vitro dissolution profiles (Figure 6A,B). These two parameters denote the cumulative percentage of curcumin dissolved at 10 min and at 120 min, respectively.

From the results shown in Table 2, it is evident that curcumin showed pH-dependent dissolution rates. Being a weak and acidic drug, the dissolution rates of curcumin and curcumin: cyclodextrin, PM, and Coppt in the simulated gastric fluid was significantly (*p* < 0.05) lower than that from the simulated intestinal fluid. For example, both the Q_10min_ and Q_120min_ estimated for curcumin at pH 1.2 was 4% and 10%; whereas, at pH 6.8, Q_10min_ and Q_120min_ for curcumin were recorded as 10% and 28%, respectively. Similarly, Q_120min_ values for curcumin: HE β-CD Coppt were 26% and 70% at pH 1.2 and pH 6.8, respectively. This marked enhancement in the dissolution rates (from an approximately 2.5 to 2.7-fold increase) was due to the pH-dependent solubility of curcumin and greater ionization of the phenolic groups at pH 6.8 [13].

On the bright side, both HE β-CD and HP β-CD Coppt showed fast and rapid initial dissolution rates in the simulated gastric fluid compared to curcumin; their Q_10min_ values improved by 5.75- and 3.75-fold, respectively, compared to that recorded for curcumin. The dispersion of HE β-CD and HP β-CD Coppt in the inclusion complexes resulted in greater solubility and better wettability of the powder due to the hydrophilic exterior of cyclodextrins, which could explain the significant enhancement in the dissolution of curcumin from HE β-CD and HP β-CD complexes. The method of preparation was also significantly (*p* < 0.05) affected by the dissolution rates of curcumin. For example, the acid Q_10min_ values for curcumin: HP β-CD PM and Coppt were 5.5% and 15%, respectively. Similarly, the acid Q_10min_ values for curcumin: HE β-CD PM and Coppt were 6.5% and 23%, respectively. The coprecipitation technique was more effective than physical mixing for the preparation of curcumin: cyclodextrin inclusion complexes. The dissolution of curcumin and cyclodextrin separately in the organic solvent and water, followed by the evaporation of the solvent system, was likely to enhance complexation efficiency and reduce particle sizes compared to physical mixing. These results correlated well with the solubility data, as abovementioned in Figure 2. The solvent evaporation was superior compared to grinding and freeze-drying techniques for the preparation of curcumin cyclodextrin complexes [24].

### 3.7. Cytotoxicity Assay Using Human Breast MCF-7 Cell Lines

Serial dilutions of the stock solution (100 µg/mL) of curcumin or its equivalent of curcumin: HE β-CD and HP β-CD Coppt were made to obtain six working concentrations in a range from 3.125 to 100 µg/mL. The results of the cytotoxicity assay on MCF-7 human breast epithelial cell lines are shown in Figure 7. Marked cell death (100%-70%) was recorded in concentrations that ranged from 25 to 100 µg/ mL for all the tested samples (Figure 7). However, the other three lower concentrations (3.125, 6.250, and 12.5 µg/mL) showed a significant reduction in cytotoxicity % especially for HP β-CD Coppt, compared to that recorded for both HE β-CD Coppt and free curcumin. This was evident from the estimated IC_50%_ for HE β-CD and HP β-CD Coppt and free curcumin, which was 7.28 ± 0.1, 19.05 ± 0.24, and 7.33 ± 0.03 µg/mL, respectively. While there were no significant (*p* > 0.050) differences between the IC_50%_ recorded for curcumin and HE β-CD Coppt, the IC_50%_ recorded for HP β-CD Coppt was significantly lower by up to 2.6-fold. These results can be explained on the ground that the capacity of cyclodextrins to reduce the cytotoxicity was due to either reducing the cytotoxic sites in the curcumin moieties and/or protecting the cell line from abrupt high local concentrations of the free drug [28,29]. Cyclodextrins are highly hydrophilic carriers with bulky structures and cannot be absorbed orally [15,16]. Curcumin is a highly lipophilic drug; therefore, curcumin is likely to leave the cyclodextrin cavities and continue the journey into systemic circulation through the portal entry/oral absorption sites by dissociating from cyclodextrin.

## 4. Conclusions

Curcumin remains one the most investigated phytochemical compounds due to its vast therapeutic potential and the interest in solving its biopharmaceutic-related problems such as solubility and permeability. In this study, β-cyclodextrin and three derivatives: Captisol, hydroxypropyl β-cyclodextrin, and hydroxyethyl β-cyclodextrins, were investigated for the formation of inclusion complexes in a 1:1 molar ratio with curcumin and were prepared using the physical mixing and solvent evaporation methods. The solvent evaporation method seemed to show better complexation efficiency, as evident from its better solubility and dissolution rates, compared to the physical mixing of curcumin and cyclodextrins. Both hydroxypropyl β-cyclodextrin and hydroxyethyl β-cyclodextrins demonstrated a superior enhancement to the solubility and dissolution rates of curcumin in both simulated gastric and intestinal fluids. However, only curcumin: hydroxyethyl β-cyclodextrins showed comparable IC_50%_ to curcumin. Future research will look into permeability improvement based on the cell uptake of curcumin from the optimized inclusion complexes using the immunofluorescence technique.

## Figures and Tables

**Figure 1 pharmaceutics-14-02283-f001:**
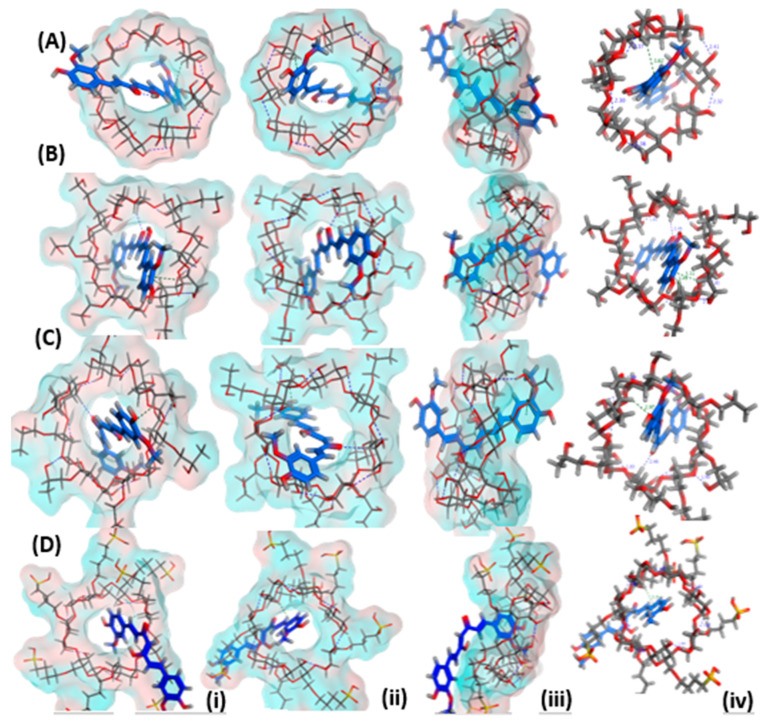
3D poses for curcumin docked into the inclusion pocket of (**A**) β-cyclodextrin, (**B**) hydroxyethyl-β-cyclodextrin, (**C**) hydroxypropyl-β-cyclodextrin and (**D**) sulfobutylether-β-cyclodextrin showing (i) top view, (ii) bottom view (iii) side view (iv) top view with marked interactions demonstrating bond length.

**Figure 2 pharmaceutics-14-02283-f002:**
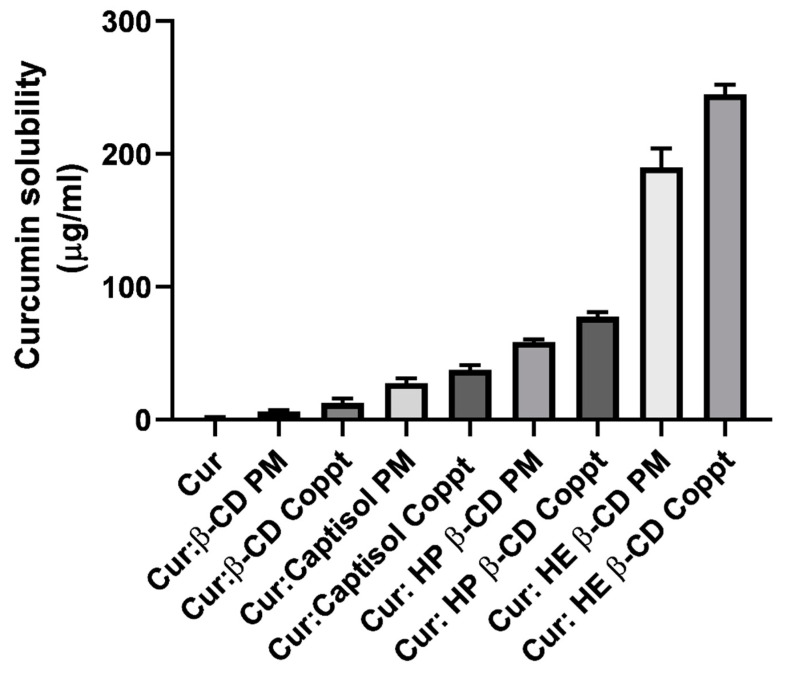
Equilibrium solubility of curcumin, curcumin: β-cyclodextrin, curcumin: Captisol, curcumin: hydroxypropyl β-cyclodextrin, and curcumin: hydroxyethyl β-cyclodextrin prepared by physical mixing (PM) and copreciptiation (Coppt), results presented as mean ± SD, *n* = 3.

**Figure 3 pharmaceutics-14-02283-f003:**
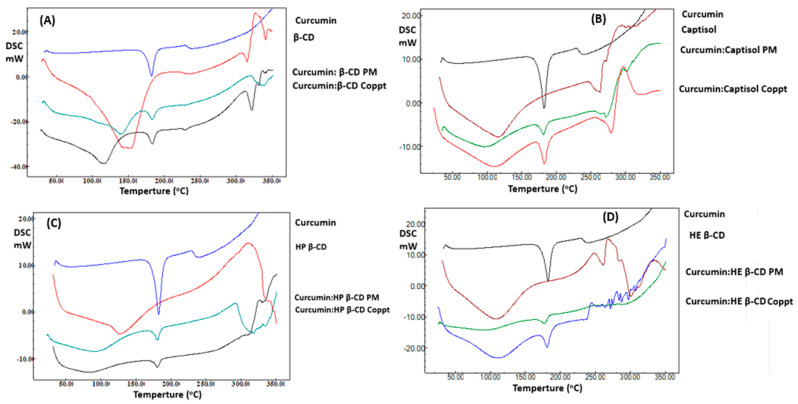
**A–D**. DSC thermograms of curcumin, curcumin: cyclodextrin complexes β-cyclodextrin (**A**), Captisol (**B**), hydroxypropyl β-cyclodextrin (**C**), and hydroxyethyl β-cyclodextrin (**D**) prepared by physical mixing (PM) and copreciptiation (Coppt).

**Figure 4 pharmaceutics-14-02283-f004:**
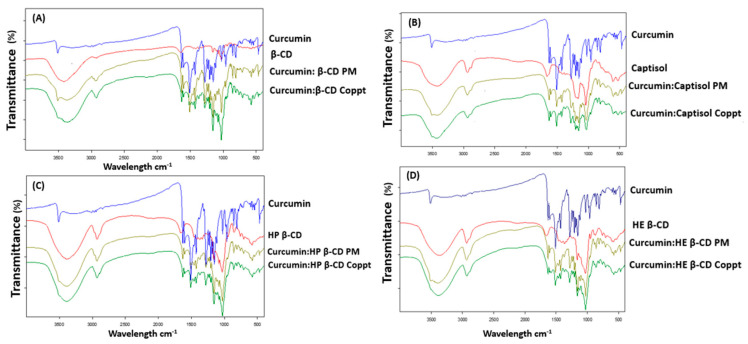
**A–D.** FTIR spectra of curcumin, curcumin: cyclodextrin complexes β-cyclodextrin (**A**), Captisol (**B**), hydroxypropyl β-cyclodextrin (**C**), and hydroxyethyl β-cyclodextrin (**D**) prepared by physical mixing (PM) and copreciptiation (Coppt).

**Figure 5 pharmaceutics-14-02283-f005:**
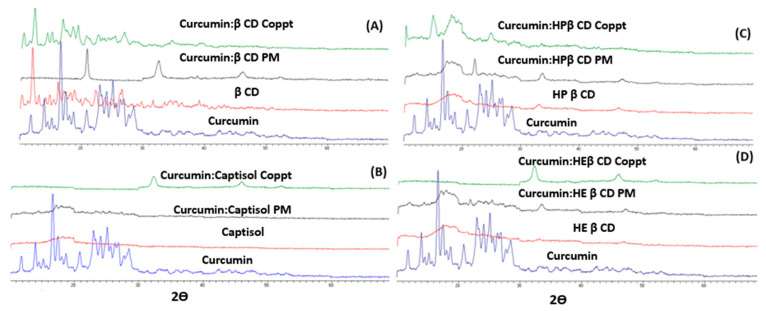
**A–D.** XRD spectra of curcumin, curcumin: cyclodextrin complexes β-cyclodextrin (**A**), Captisol (**B**), hydroxypropyl β-cyclodextrin (**C**), and hydroxyethyl β-cyclodextrin (**D**) prepared by physical mixing (PM) and copreciptiation (Coppt).

**Figure 6 pharmaceutics-14-02283-f006:**
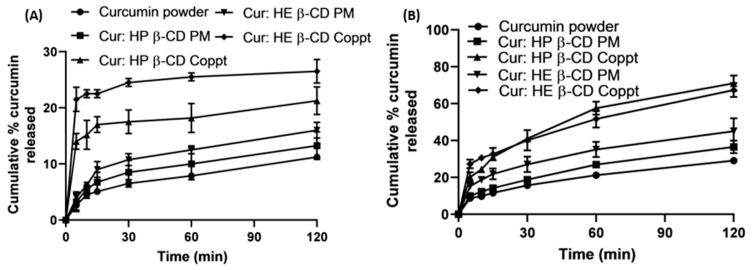
In vitro curcumin dissolution from curcumin alone and some selected physical (PM) and coprecipitated (Coppt) with HP β-CD and HE β-CD in simulated gastric fluid (**A**) and simulated intestinal fluid (**B**). Results presented as mean ± SD, *n* = 3.

**Figure 7 pharmaceutics-14-02283-f007:**
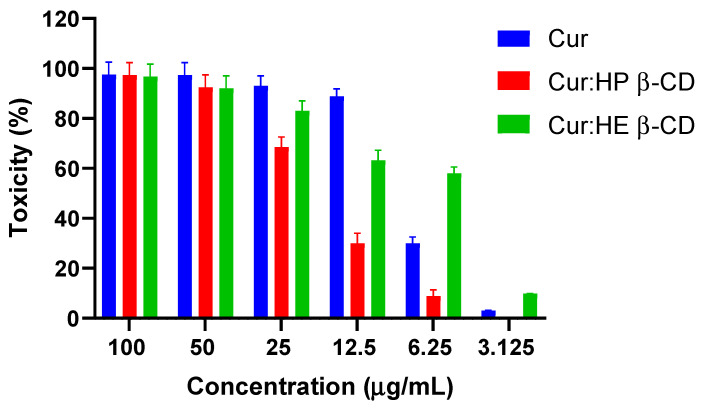
Cytotoxicity (%) estimated for six serially diluted concentrations (3.125 to 100 µg/mL) curcumin, curcumin: HP β-CD Coppt, curcumin:HE β-CD Coppt. Results expressed as mean ± SD, *n* = 3.

**Table 1 pharmaceutics-14-02283-t001:** Energy score (kcal/mol) and number of potential interactions observed for Curcumin docked into the inclusion pocket of β-cyclodextrin, Hydroxyethyl-β-cyclodextrin, Hydroxypropyl-β-cyclodextrin, and sulfobutylether-β-cyclodextrin.

Drug	Carrier	E Score (kcal/mol)	Number of Interactions	
			H-Bond	Hydrophobic Interactions
Curcumin	β-cyclodextrin	−5.2617	-	1
	Hydroxyethyl-β-cyclodextrin	−6.0093	1	2
	Hydroxypropyl-β-cyclodextrin	−5.8415	1	1
	Sulfobutylether-β-cyclodextrin	−5.7392	-	1

**Table 2 pharmaceutics-14-02283-t002:** In vitro dissolution parameters (Q_10min_ and Q_120min_) estimated for curcumin and some selected curcumin: cyclodextrins, PM, and Coppt. Results represent mean ± SD, *n* = 3.

Formulation/Preparation Method	Simulated Gastric Fluid (pH 1.2)		Simulated Intestinal Fluid (pH 6.8)	
	Q_10min_ (%)	Q_120min_ (%)	Q_10min_ (%)	Q_120min_ (%)
Curcumin	4 ± 0.5	10 ± 1.0	10 ± 0.5	28 ± 3.5
Curcumin: HP β-CD PM	5.5± 1.0	13 ± 1.0	12 ± 0.5	36 ± 3.0
Curcumin: HP β-CD Coppt	15 ± 2.5	21 ± 2.0	25 ± 3.0	72 ± 4.0
Curcumin: HE β-CD PM	6.5 ± 1.0	15 ± 1.0	18 ± 2.0	45 ± 3.0
Curcumin: HE β-CD Coppt	23 ± 4.0	26 ± 3.5	30 ± 3.5	70 ± 4.0

## Data Availability

Upon request.

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
