# Peer review of "Design, Preparation and Evaluation of Supramolecular Complexes with Curcumin for Enhanced Cytotoxicity in Breast Cancer Cell Lines"

_pharmaceutics, 2022, doi:10.3390/pharmaceutics14112283_

Round 1

Reviewer 1 Report

In this study, the Authors show that some different cyclodextrin derivatives (the native β-cyclodextrin, sufobutyl ether β-CD, hydroxypropyl β-cyclodextrin and hydroxylethyl β-cyclodextrin) associated with curcumin are able to enhance both solubility and anti- cancer activities in human breast cancer cell lines. To prepare curcumin-cyclodextrins complexes, two different techniques (physical mixing and coprecipitation method) were used. The results are interesting, but in the present form, the paper cannot be accepted for the publication, since there are several problems which require a major and careful revision as suggested below.

-To improve the paper quality, I suggest to considerer an in vitro cellular uptake study by immunofluorescence technique.

-Materials and Methods section: pags. 3-4 lines 140-144 in In vitro dissolution studies, the authors describe two dissolution media: ”the first dissolution medium was composed of simulated gastric fluid ……………. The second dissolution medium was composed of phosphate buffer with pH 6.8……..”. This sentence should be correct, as the media described in Results 3.2 are gastric and intestinal fluids (see Fig. 7 and table 2).

-Fig. 1 is not mentioned in the text (results 3.1)

-Pag.7 line 236: Fig.3A-D should be changed in Fig. 4 A-D.

-To get an overview, I suggest to assemble figures 5 and 6.

-Pag.10 lines 277-299: this paragraph should be re-written more clearly and carefully.

-Fig.7: I suggest to add “Gastic Fluid” in A and “Intestinal Fluid” in B.

In addition, throughout the text some grammatical and typing errors should be revised and corrected.

Author Response

Reviewers 1:

In this study, the Authors show that some different cyclodextrin derivatives (the native β-cyclodextrin, sufobutyl ether β-CD, hydroxypropyl β-cyclodextrin and hydroxylethyl β-cyclodextrin) associated with curcumin are able to enhance both solubility and anti- cancer activities in human breast cancer cell lines. To prepare curcumin-cyclodextrins complexes, two different techniques (physical mixing and coprecipitation method) were used. The results are interesting, but in the present form, the paper cannot be accepted for the publication, since there are several problems which require a major and careful revision as suggested below.

-To improve the paper quality, I suggest to considerer an in vitro cellular uptake study by immunofluorescence technique.

This a valid point and due time constraints we reserved for future study. This has now been highlighted in the conclusion section as directions for future research.

-Materials and Methods section: pags. 3-4 lines 140-144 in In vitro dissolution studies, the authors describe two dissolution media: ”the first dissolution medium was composed of simulated gastric fluid ……………. The second dissolution medium was composed of phosphate buffer with pH 6.8……..”. This sentence should be correct, as the media described in Results 3.2 are gastric and intestinal fluids (see Fig. 7 and table 2).

The section has been modified accordingly. Figure 7’ caption has been also corrected.

-Fig. 1 is not mentioned in the text (results 3.1).

Figure 1 has nowbeen mentioned in the text.

-Pag.7 line 236: Fig.3A-D should be changed in Fig. 4 A-D.

The figure number has now been corrected.

-To get an overview, I suggest to assemble figures 5 and 6.

-Pag.10 lines 277-299: this paragraph should be re-written more clearly and carefully.

The whole paragraph has now been revised and rewritten.

-Fig.7: I suggest to add “Gastic Fluid” in A and “Intestinal Fluid” in B.

In addition, throughout the text some grammatical and typing errors should be revised and corrected.

The whole manuscript has now been revised for English and proofreading.

Reviewer 2 Report

The Authors describe interdisciplinary research on composite materials, supramolecular complexes with curcumin. The manuscript includes their synthesis, computational pharmacological analysis, in vitro biological research. However there are some important points that the authors have to improve:

1)      In the introduction, you should find data from the literature related to the host matrix, cyclodextrin and the stage of the studies of curcumin encapsulation in this matrix.

2)      What purity do the raw materials have? It is required due to biological applications (cytotoxicity in vitro).

3)      In fig.4, FTIR spectra are not good. FTIR spectra are not made in absorbance, please correct and indicate which curve corresponds to which sample by other method, not the color of curves.

Indicate the characteristic peaks of each compound and highlight the differences between the raw materials and the composite materials obtained.

4)      In Fig. 5 and 6, please mark the main diffraction peaks, and assign them accurately.

5)      The graph in figure 8 must contain the control bars, for a better interpretation of the same.

6)      The authors must specify which of the two methods of obtaining the synthesized materials is better and for what reason they assume it is so.

7)      Bibliographic references must be reviewed

References should be described as follows, depending on the type of work:

  Journal Articles:
1. Author 1, A.B.; Author 2, C.D. Title of the article. Abbreviated Journal Name YearVolume, page range.

Author Response

Reviewer 2:

The Authors describe interdisciplinary research on composite materials, supramolecular complexes with curcumin. The manuscript includes their synthesis, computational pharmacological analysis, in vitro biological research. However there are some important points that the authors have to improve:

  • In the introduction, you should find data from the literature related to the host matrix, cyclodextrin and the stage of the studies of curcumin encapsulation in this matrix.

The introduction has now been modified accordingly.

2)      What purity do the raw materials have? It is required due to biological applications (cytotoxicity in vitro).

The purity has now been provided (Section 2.1.)

3)      In fig.4, FTIR spectra are not good. FTIR spectra are not made in absorbance, please correct and indicate which curve corresponds to which sample by other method, not the color of curves.

Indicate the characteristic peaks of each compound and highlight the differences between the raw materials and the composite materials obtained.

4)      In Fig. 5 and 6, please mark the main diffraction peaks, and assign them accurately.

The characteristics diffraction peaks have been  assigned accordingly.

5)      The graph in figure 8 must contain the control bars, for a better interpretation of the same.

The negative control was the media and it did not show any sing of toxicity, so the control read 0 % (insignificant cytotoxicity).

6)      The authors must specify which of the two methods of obtaining the synthesized materials is better and for what reason they assume it is so.

This has now been clearly mentioned in the conclusion section.

7)      Bibliographic references must be reviewed

References should be described as follows, depending on the type of work:

  Journal Articles:

  1. Author 1, A.B.; Author 2, C.D. Title of the article. Abbreviated Journal Name Year, Volume, page range.

The reference list has now been modified accordingly.

Round 2

Reviewer 1 Report

The paper results improved and could be accepted with some minor revision, as indicated below:

-       - In my previous comments I suggested to re-written more clearly the paragraph placed in Pag.10 lines 277-299. In the following sentence: “For example, both Q10min and  Q120min estimated for curcumin at pH 1.2 was 4% and 23% while at pH 6.8 recorded 10%  and 28%, respectively” I think that 23% should be replace with 10%, as reported in the table 2.

-          - Reference 17: The Authors should write the year 2022.

Author Response

Reviewers 1:

The paper results improved and could be accepted with some minor revision, as indicated below:

-       - In my previous comments I suggested to re-written more clearly the paragraph placed in Pag.10 lines 277-299. In the following sentence: “For example, both Q10min and  Q120min estimated for curcumin at pH 1.2 was 4% and 23% while at pH 6.8 recorded 10%  and 28%, respectively” I think that 23% should be replace with 10%, as reported in the table 2.

This is a valid point. The percentage has been corrected and the whole paragraph was revised.

-          - Reference 17: The Authors should write the year 2022.

The year has now been corrected.

Reviewer 2 Report

Please, in fig. 4, complete with (%) after Transmittance and delete numerical values on the ordinates as these are arbitrary values and have no significant meaning.

Line 249, correct please, Van der Waals.

Given the above, I am of opinion that this paper should be accepted after the suggested corrections.

Author Response

Reviewer 2:

Please, in fig. 4, complete with (%) after Transmittance and delete numerical values on the ordinates as these are arbitrary values and have no significant meaning.

The figure has now been modified accordingly.

Line 249, correct please, Van der Waals.

The name has been corrected.

Given the above, I am of opinion that this paper should be accepted after the suggested corrections.